# Performance Evaluations of Distributed File Systems for Scientific Big Data in FUSE Environment

**Jun-Yeong Lee** [1], **Moon-Hyun Kim** [1], **Syed Asif Raza Shah** [2], **Sang-Un Ahn** [3] **and Heejun Yoon** [3] **and Seo-Young Noh** [1,*]

1   Department of Computer Science, Chungbuk National University, Cheongju-si 28644, Korea; lee1238234@cbnu.ac.kr (J.-Y.L.); moonhyunkim@cbnu.ac.kr (M.-H.K.)
2   Department of Computer Science and CRAIB, Sukkur IBA University (SIBAU), Sukkur 65200, Pakistan; asif.shah@iba-suk.edu.pk
3   Global Science Experimental Data Hub Center, Korea Institute of Science and Technology Information, 245 Daehak-ro, Yuseong-gu, Daejeon 34141, Korea; sahn@kisti.re.kr (S.-U.A.); k2@kisti.re.kr (H.Y.)
*   Correspondence: rsyoung@cbnu.ac.kr

**Abstract:** Data are important and ever growing in data-intensive scientific environments. Such research data growth requires data storage systems that play pivotal roles in data management and analysis for scientific discoveries. Redundant Array of Independent Disks (RAID), a well-known storage technology combining multiple disks into a single large logical volume, has been widely used for the purpose of data redundancy and performance improvement. However, this requires RAID-capable hardware or software to build up a RAID-enabled disk array. In addition, it is difficult to scale up the RAID-based storage. In order to mitigate such a problem, many distributed file systems have been developed and are being actively used in various environments, especially in data-intensive computing facilities, where a tremendous amount of data have to be handled. In this study, we investigated and benchmarked various distributed file systems, such as Ceph, GlusterFS, Lustre and EOS for data-intensive environments. In our experiment, we configured the distributed file systems under a Reliable Array of Independent Nodes (RAIN) structure and a Filesystem in Userspace (FUSE) environment. Our results identify the characteristics of each file system that affect the read and write performance depending on the features of data, which have to be considered in data-intensive computing environments.

**Keywords:** data-intensive computing; distributed file system; RAIN; FUSE; Ceph; EOS; GlusterFS; Lustre

## 1. Introduction

As the amount of computing data increases, the importance of data storage is emerging. Research from IDC and Seagate predicted that the size of the global data sphere was only a few ZB in 2010, but it would increase to 175 ZB by 2025 [1]. CERN, one of the largest physics research groups in the world, produces 125 petabytes of data per year from LHC experiments [2]. Due to the tremendous amount of experimental data produced, data storage is one of key factors in scientific computing. In such a computing environment, the capacity and stability of storage systems are important because the speed of data generation is high, and it is almost impossible to reproduce the data. Although there are many approaches to handling such big data, RAID has been commonly used to store large amounts of data because of its reliability and safety. However, RAID requires specific hardware and software to configure or modify storage systems. Moreover, it is difficult to expand with additional storage capacity if it is predefined. In addition, rebuilding a RAID is likely to affect the stability of the RAID system, which may result in total data loss. To overcome these drawbacks, many distributed file systems have been developed and deployed at many computing facilities for data-intensive research institutes. The

distributed file system provides horizontal scalability compared to RAID, which uses vertical scalability. Additionally, some distributed file systems provide geo-replication, allowing data to be geographically replicated throughout the sites. Due to these features, distributed file systems provide more redundancy than RAID storage systems. Distributed file systems are widely deployed at many data-intensive computing facilities. EOS, one of the distributed file systems, was developed by CERN in 2010. It is currently deployed for storing approximately 340 petabytes, consisting of 6 billion files [3]. Many national laboratories and supercomputing centers, like Oak Ridge National Laboratory, use Lustre for their storage for high-performance computing [4]. In this study, we deployed and evaluated numerous distributed file systems using a small cluster with inexpensive server hardware and analyzed the performance characteristics for each file system. We configured a RAID 6-like RAIN data storing system and distributed data storing systems and measured the performance of file systems by accessing data using a FUSE client rather than using vendor-specific APIs and benchmarking tools. Our approach can allow us to distinguish the main performance differences of distributed file systems in userspace which are directly affecting user experiences. Our experimental results show that the performance impacts depend on the scientific data analysis scenarios. Therefore, it is expected that the outcomes of our research can provide valuable insights which can help scientists when deploying distributed file systems in their data-intensive computing environments, considering the characteristics of their data.

The rest of this paper is organized as follows: In Section 2, we describe which technologies and distributed file systems were used for our evaluation. In Section 3, we describe previous studies relevant to our research. In Section 4, we describe our evaluation environment and configuration of hardware and software which were used for our evaluation. In Sections 5 and 6, we cover the results of our evaluation. Finally, Section 7 describes our conclusions about the results and future plans.

## 2. Backgrounds

In this section, we discuss the background knowledge related to our work, such as RAIN.

### 2.1. RAIN

Reliable Array of Independent Nodes (RAIN) is a collaboration project from the Caltech Parallel and Distributed Computing Group and Jet Propulsion Laboratory from NASA [5]. The purpose of this project was to create a reliable parallel computing cluster using commodity hardware and storage with multiple network connections. RAIN implements redundancy using multiple computing nodes and storage nodes which consist of heterogeneous clusters. RAIN features scalability, dynamic reconfiguration and high availability. RAIN can handle failures using four main techniques, described below:

- Implement multiple network interfaces at nodes.
- Single point of failure prevention using network monitoring.
- Cluster monitoring using grouping.
- Storage node redundancy using error-correcting code such as RAID.

### 2.2. FUSE

Filesystem in Userspace (FUSE) [6] is an interface library that passes a file system to a Linux kernel from the userspace program included in most Linux distributions. Implementing a file system directly through the Linux kernel is very difficult, but using the FUSE library allows a file system to be configured without manipulating the kernel directly. FUSE provides high-level and low-level API, and supports various platforms like Linux, BSD and MacOS. Due to these characteristics, hundreds of file systems have been implemented using the FUSE library [7].

### 2.3. Ceph

Ceph [8] is an open-source distributed file system developed by the University of California and maintained by the Ceph Foundation. This file system provides object, block and file storage in a unified system. In addition, it uses the Reliable Autonomous Distributed Object Store (RADOS) to provide reliable and high-performance storage that can scale up from petabyte to exabyte capacity. RADOS consists of a monitor (MON), manager (MGR), object storage daemon (OSD) and metadata server (MDS). MON maintains a master copy of the cluster map, which contains the topology of the cluster. MGR runs with MON, which provides an additional monitoring interface for external monitoring and management systems. OSD interacts with logical disk and handles data read, write and replicate operations on actual physical disk drives. MDS provides metadata to CephFS for serving file services. Ceph stores data using the Controlled Replication Under Scalable Hashing (CRUSH) algorithm, which can place and locate data using the hash algorithm [9]. Figure 1 shows the structure of Ceph.

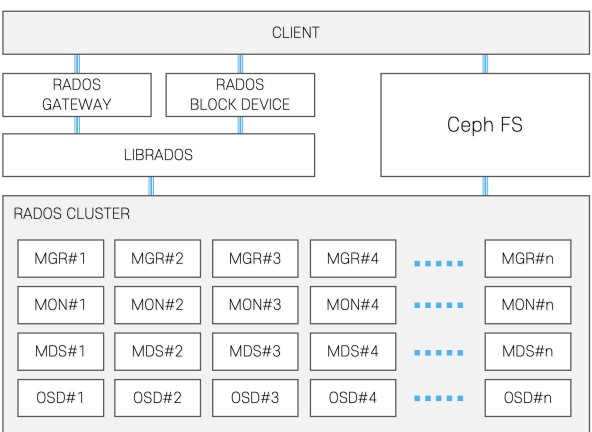

**Figure 1.** Structure of Ceph.

### 2.4. EOS

EOS [10] is an open-source disk-based distributed storage developed by CERN. It is used to store LHC experiment data and user data at CERN. EOS natively supports the XRootD protocol, but also supports various other protocols, such as HTTP, WebDAV, GridFTP and FUSE. EOS consists of three components: MGM, FST and MQ. Figure 2 shows the structure of EOS. MGM is a management server that manages the namespace, file system quota and file placement and location. FST is a file storage server that stores data and metadata. MQ is a message queue that provides asynchronous messaging between MGM and FST. EOS uses the "layout" to store data [11]. The layout determines how data can be stored on the file system. Some layouts can support duplication or erasure coding which can prevent data loss and accidents. The layouts are shown in Table 1.

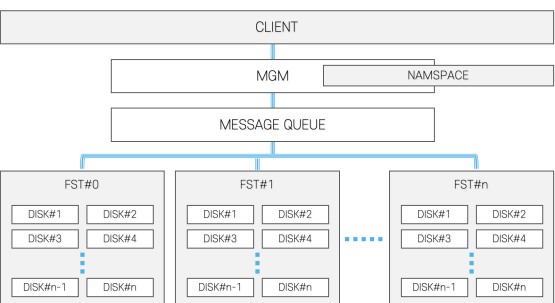

**Figure 2.** Structure of EOS.

**Table 1.** EOS layouts [11].

| Name | Redundancy | Algorithm |
| --- | --- | --- |
| plain | None | Stores data on one file system |
| replica | N | Create N replicas and stores on N file systems |
| raid5 | N+1 | Single parity RAID |
| raiddp | 4+2 | Dual parity RAID |
| raid6 | N+2 | Erasure code (Jerasure Library) |
| archive | N+3 | Erasure code (Jerasure Library) |
| qrain | N+4 | Erasure code (Jerasure Library) |

### 2.5. GlusterFS

GlusterFS is an open-source distributed file system that is developed and supported by RedHat [12]. This file system binds multiple server disk resources into a single global namespace using the network. GlusterFS can be scaled up to several petabytes and can be used with commodity hardware to create storage. GlusterFS provides replication, quotas, geo-replication, snapshots and bit rot detection. Unlike other distributed file systems, GlusterFS has no central management node or metadata node. GlusterFS can be accessed not only with the GlusterFS native client, but can also be accessed with various protocols, such as a network file system (NFS), service message block (SMB) and common interest file system (CIFS). Figure 3 shows the architecture of GlusterFS.

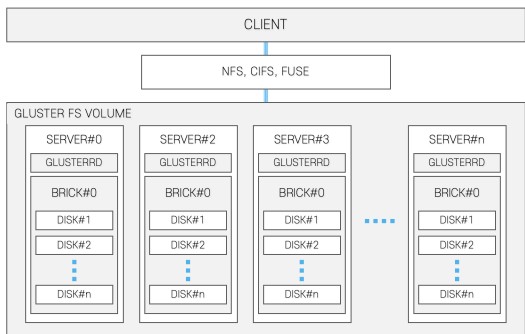

**Figure 3.** Structure of GlusterFS.

GlusterFS stores data in a place called a volume that consists of multiple bricks, which can be a single disk or a just a bunch of disks (JBOD) enclosure [13]. The volume supports different types of stored data, and some types of volume support duplication or erasure coding. Table 2 shows the types of volume used in GlusterFS.

**Table 2.** GlusterFS volumes.

| Name | Redundancy | Algorithm |
| --- | --- | --- |
| Distributed | None | Stores data on one file system |
| Replicated | N | Create N replicas and stores on N bricks |
| Distributed Replicated | N*Count | Distributed across replicated sets of volumes |
| Dispersed | N+K | Erasure coding with K redundancy |

### 2.6. Lustre

Lustre is an open-source distributed file system designed for high-performance computing [4]. Lustre started from Carnegie Mellon University's project and it is currently used in many high-performance computing clusters. It uses distributed object storage architecture [14], which consists of a management server, metadata server and object storage server. The management server (MGS) manages all Lustre servers and clients. In addition, it stores the server configuration. The metadata server (MDS) stores metadata information. Multiple metadata servers can be deployed to scale up metadata storage and provide more

redundancy. The object storage server (OSS) provides the storage for data. It uses striping to maximize performance and storage capacity. Figure 4 shows the architecture of Lustre.

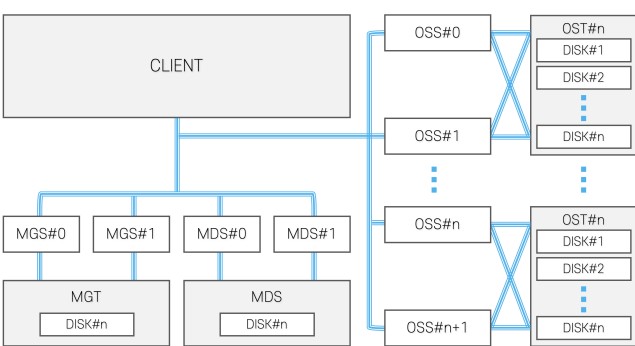

**Figure 4.** Structure of Lustre.

## 3. Related Work

There are several studies which have been conducted to evaluate the performance of distributed file systems.

Diana et al. [15] implemented Ceph using commodity servers to provide multi-use, highly available and performance-efficient file storage for a variety of applications, from shared home directories to the scratch directories of high-performance computing. They evaluated scalability for Ceph by increasing the object storage server, number of clients and object size to understand which factors affect file system performance. They benchmarked Ceph using `rados bench` and RADOS block device with `fio`. In their experiment, cluster network performance was measured using `netcat` and `iperf` while the individual data disk performance was measured using `osd tell` to make a baseline for file system performance.

Zhang et al. [16] virtually deployed a Ceph cluster in an OpenStack environment to evaluate the performance of Ceph deployed in the virtual environment. They benchmarked the cluster's network performance using `netcat` and `iperf`. They used `rados bench` and RADOS block device with `bonnie++` to measure the performance.

Kumar [17] configured GlusterFS in a software-defined network (SDN) environment with six remote servers and analyzed how GlusterFS performs in large-scale scientific applications. Through the environment, they evaluated GlusterFS and network performance. With the evaluation result, they proposed which kind of quality of service (QoS) policy has to provide for certain users for servicing GlusterFS in federated cloud environments.

Luca et al. [18] presented different distributed file systems used in modern cloud services, including HDFS, Ceph, GlusterFS and XtremeFS. They focused on writing performance, fault tolerance and re-balancing ability for each file system. They also evaluated deployment time for each distributed file system.

In addition, several benchmark tools designed to evaluate distributed file systems have also been introduced. Xin Li et al. [19] developed the LZpack benchmark for distributed file systems that can test metadata and file I/O performance and evaluated file system performance using Lustre and NFS.

Jaemyoun Lee et al. [20] proposed a large-scale object storage benchmark based on the Yahoo! Cloud Serving Benchmark (YCSB) [21]. They developed a YCSB client for Ceph RADOS, which can communicate between Ceph and YCSB to evaluate the performance of Ceph storage using the YCSB.

Although there are many methods to evaluate the distributed file system performance, two approaches are mainly used when evaluating the performance of file systems. The first is using the file system's own tool, for example, `rados bench` of Ceph [22] and TestDFSIO of Hadoop [23]. The second is mounting the file system using the FUSE client and benchmark using various tools such as `dd` [24], `bonnie++` [25], `iozone` [26] and `fio` [27]. The first method can verify performance under specific file systems, but the other file systems cannot use the file system's API or tools to find performance differences using file system-

specific or biased tools. However, if we use FUSE clients from each file system, we can mount the file systems in same Linux userspace and verify performance using the same tools with the same parameters. Therefore, it is possible to evaluate the distributed file systems with the same conditions, resulting in fair performance comparisons, which can give valuable insights to scientists when adapting distributed file systems in their data-intensive computing environments.

We can find various studies [15–17] which have measured the performance of file systems using various tools. Other papers [19,20] also describe their own tools to benchmark the file system. However, it is not easy to find research papers describing FUSE clients to evaluate the performance of distributed file systems.

In this study, we evaluated the storage performance using FUSE clients provided by each distributed file system with the FIO benchmark. We selected this method because using FUSE clients can evaluate file systems with identical parameters, which is important for a fair comparison among file systems.

## 4. The Experimental Setup

In this section, we describe the environment setup for our experiment and the way we evaluated the performance for each file system.

### 4.1. Hardware

To benchmark the distributed file system, we configured a small cluster environment for simulating a small distributed system. Our testing cluster environment had four servers for deploying and testing the distributed file system. We configured the master server to act as a management node to test the file systems. We also set up three slave servers to act as storage for the distributed file systems. The detailed specifications for all servers are listed in Table 3. All slave servers were configured identically to minimize variables during the evaluation. For OS, we installed CentOS 7.8.2003 Minimal on all servers. Then we separated the boot disk and data disk of all servers to prevent interference between OS and storage for the distributed file systems.

**Table 3.** Server specification.

|  | Master Server | Slave Server |
| --- | --- | --- |
| Chassis | HP ProLiant ML350 G9 | HP ProLiant ML350 G6 |
| CPU | E5-2609v3 1.90 GHz | E5606 2.13 GHz |
| RAM | 40 GB | 4 GB |
| Disk | 120 GB SSD (Boot) | 500 GB HDD (Boot) |
|  | 2 * 1 TB HDD (Data) | 2 * 1 TB HDD (Data) |
| Network | 3 Gbit (1 Gbit * 3) | 1 Gbit |

### 4.2. Network

Figure 5 shows the overall network configuration for our cluster. We used network bonding using `nmtui` to bond three 1 Gbit network interfaces to create a single 3 Gbit logical interface in the master server. In this way, we could minimize the bottleneck between the three slave servers.

Slave servers were configured with a single 1 Gbit network interface and connected to the same router as the master server. To evaluate the network configuration, `iperf` benchmark was performed simultaneously between slave servers and the master server. Table 4 shows the `iperf` results from our cluster.

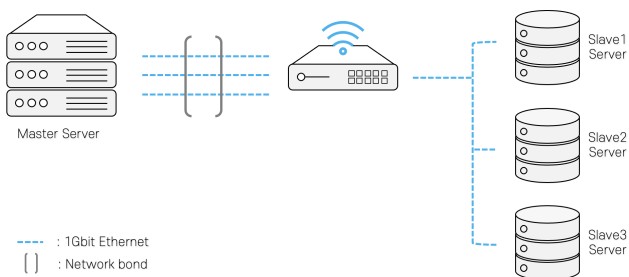

**Figure 5.** Cluster network configuration.

**Table 4.** `iperf` benchmark between slave servers and master server.

| Bandwidth | Slave1 | Slave2 | Slave3 |
|---|---|---|---|
| Mbit/s | 910 | 914 | 913 |

The bandwidth result of each slave server was about 910 Mbit/s due to the router's internal hardware traffic limit, summed up as 2730 Mbit/s.

### 4.3. Distributed File System

We set up each distributed file system on the cluster as we described in the "Hardware" subsection. After evaluating each distributed file system, we re-initialized all the boot and data disks in order to prevent the next evaluation from being affected by an unflushed cache and data from the previous test. Therefore, all experiments were conducted under the same conditions. All file systems were mounted on the system via provided FUSE clients to evaluate the performance.

#### 4.3.1. Environment 1: Ceph

We installed Ceph Mimic 13.2.10 on all servers. MGR and MON services were installed on the master server to manage and monitor the Ceph cluster. The data disks of slave servers were initialized with a default Bluestore storage backend to configure Ceph storage. We configured CephFS with metadata and data pools and mounted it on the master server using FUSE clients. Figure 6 shows our configured environment for Ceph, where MON and MGR services are configured and running on the master server, and 6 OSD services are configured on slave servers.

```
[root@master ~]# ceph -s
  cluster:
    id:     ffea44f4-6226-47e8-9024-8be9b14d6da5
    health: HEALTH_OK

  services:
    mon: 1 daemons, quorum master
    mgr: master.dclab(active)
    osd: 6 osds: 6 up, 6 in

  data:
    pools:   2 pools, 256 pgs
    objects: 0  objects, 0 B
    usage:   6.1 GiB used, 5.5 TiB / 5.5 TiB avail
    pgs:     256 active+clean
```

**Figure 6.** Ceph cluster status.

#### 4.3.2. Environment 2: EOS

We installed EOS 4.7.5 on all servers. On the master server, MGM and MQ services were installed to manage metadata and asynchronous messaging between MGM and FST. The data disks of slave servers were initialized to the XFS file system and mounted on the system. From the master server, we registered data disks from slave servers to create default space for EOS storage. The created default space was mounted on the master using FUSE clients. Figure 7 shows our environment status on the EOS console. The console shows registered data disks and servers for EOS storage.

```
EOS Console [root://localhost] |/eos/> fs ls

host                    port    id                              path    schedgroup

slave1.dclab            1095    3                              /data0   default.0
slave1.dclab            1095    4                              /data1   default.0
slave2.dclab            1095    5                              /data0   default.0
slave2.dclab            1095    6                              /data1   default.0
slave3.dclab            1095    7                              /data0   default.0
slave3.dclab            1095    8                              /data1   default.0

EOS Console [root://localhost] |/eos/> node ls

type                              hostport        geotag      status    activated

nodesview              slave1.dclab:1095        EOS::TEST    online        on
nodesview              slave2.dclab:1095        EOS::TEST    online        on
nodesview              slave3.dclab:1095        EOS::TEST    online        on
```

**Figure 7.** EOS console.

### 4.3.3. Environment 3: GlusterFS

GlusterFS 7.6 was installed on all servers. Data disks on slave servers were initialized to the XFS file system and mounted on the system as a brick. We created a GlusterFS volume with initialized disks and the volume was mounted with the FUSE client on the master server. Figure 8 shows our GlusterFS volume status with registered data disks.

```
[root@master ~]# gluster volume status
Status of volume: glusterfs
Gluster process                                   TCP Port  RDMA Port  Online  Pid
------------------------------------------------------------------------------------
Brick slave1.test:/data0/brick00                   49152       0         Y      3224
Brick slave1.test:/data1/brick01                   49153       0         Y      3234
Brick slave2.test:/data2/brick02                   49154       0         Y      3243
Brick slave2.test:/data3/brick03                   49155       0         Y      3251
Brick slave3.test:/data4/brick04                   49156       0         Y      3255
Brick slave3.test:/data5/brick05                   49157       0         Y      3258
Self-heal daemon on localhost                        N/A       N/A       Y      3262

Task Status of Volume glusterfs
------------------------------------------------------------------------------------
There are no active volume tasks
```

**Figure 8.** Gluster volume status.

### 4.3.4. Environment 4: Lustre

Lustre 2.13.0 and a Lustre-specific kernel was installed on all servers. For managing the Lustre cluster, management target (MGT) and metadata target (MDT) services were configured on the master server. The data disks of slave servers were initialized to object storage target (OST). Finally, all disks were mounted on the system to start the Lustre file system. The configured Lustre file system was mounted on the master server. Figure 9 shows our configured Lustre cluster where configured MDT and OST data disks are initialized and registered on the Lustre file system.

```
[root@master ~]# lfs df -h
UUID                     bytes       Used    Available Use% Mounted on
test-MDT0000_UUID         4.4G       1.9M         4.0G   1% /root/lustre[MDT:0]
test-OST0001_UUID       916.8G       1.2M       870.2G   1% /root/lustre[OST:1]
test-OST0002_UUID       916.8G       1.2M       870.2G   1% /root/lustre[OST:2]
test-OST0003_UUID       916.8G       1.2M       870.2G   1% /root/lustre[OST:3]
test-OST0004_UUID       916.8G       1.2M       870.2G   1% /root/lustre[OST:4]
test-OST0005_UUID       916.8G       1.2M       870.2G   1% /root/lustre[OST:5]
test-OST0006_UUID       916.8G       1.2M       870.2G   1% /root/lustre[OST:6]

filesystem_summary:       5.3T       7.3M         5.1T   1% /root/lustre
```

**Figure 9.** Lustre status.

### 4.4. File System Layouts

In order to benchmark the performance of the distributed file systems as mentioned above, the layout of the file systems was required. We used two different file system layouts—"distributed layout" and "RAID 6-like RAIN layout"—to evaluate the performance and characteristics.

#### 4.4.1. Distributed Layout

The distributed layout stored data linearly across the entire disk array. It can be seen that it was very efficient because all disk space could be utilized as storage. However, there was no data redundancy, so there was a potential for critical data loss from disk failures. We tested the distributed layout on all distributed file systems. Table 5 describes the options we used in each file system.

**Table 5.** Distributed layout.

| Distributed File System | Options |
|:---:|:---:|
| Ceph | OSD Pool Size = 1 |
| EOS | Plain |
| GlusterFS | Distributed Layout |
| Lustre | Stripe = 1 |

### 4.4.2. RAID 6-like RAIN Layout

Like RAID 6, the RAIN layout used in this evaluation enabled erasure coding to calculate additional parity data for data redundancy. It could endure two disk failures. However, if there are more than two disk failures, the entire storage may fail because it cannot calculate the original data from storage clusters. We tested the RAIN layout on all distributed file systems except Lustre. Although Lustre's development roadmap has a plan for erasure-coded pools [28], they were not officially supported when we designed our benchmark. Table 6 describes the options we used in each file system.

**Table 6.** RAID 6-like RAIN Layout.

| Distributed File System | Options |
|:---:|:---:|
| Ceph | erasure-code-profile, k = 4, m = 2 |
| EOS | RAID 6, stripe = 6 |
| GlusterFS | Dispersed Volume, k = 4, m = 2 |

### 4.5. Evaluation Software and Methods

We used `fio` 3.7 [27] to measure and evaluate the performance. We automated the benchmark using Linux shell scripts. Each evaluation with the `fio` benchmark ran for 180 s using the `libaio` I/O engine. We set the block size to 4 K, 128 K and 1024 K, respectively. The benchmark was done with different I/O patterns for each block size. There were four I/O patterns in our experiment: sequential write, sequential read, random read and random write. In addition, tests were conducted by increasing the number of threads for each experimental scenario to 1, 2, 4 and 8 in order to simulate the increasing workload. Table 7 shows the options we used with `fio`.

**Table 7.** fio options.

| Options | Parameters |
|:---:|:---:|
| Block Size | 4 K, 128 K, 1024 K |
| Job Size | 1 GB |
| Number of Threads | 1, 2, 4, 8 |
| IODepth | 32 |
| Evaluated I/O Pattern | Sequential Read, Sequential Write |
| | Random Read, Random Write |

## 5. Results

In this section, we discuss the evaluation results from our experiments. The measured results are described by dividing them into layouts, and in the case of the RAID 6-like RAIN layout, the benchmark results of three distributed file systems excluding Luster are shown as graphs because Luster does not support the corresponding layout.

### 5.1. Distributed Layout

### 5.1.1. Sequential Read Throughput

Figure 10 shows the result of the sequential read benchmark. Ceph showed relatively high throughput across all block sizes. However, the 4 K and 128 K block results showed a decrease in throughput after four threads. At the 1024 K block, there was no decrease

in throughput due to increasing threads, but the increase in performance with increasing threads was minimal. EOS increases throughput without deteriorating as the number of threads increases at all block sizes. Unlike the other file systems, GlusterFS shows similar performance between one and two threads at all block sizes. For more than two threads, throughput was found to increase like any other file system. Lustre's performance was enhanced in a similar way to EOS. The 4 K and 128 K block evaluations showed low performance compared to the other distributed file systems. However, at a 1024 K block with eight threads, Luster showed the highest performance compared to the other file systems.

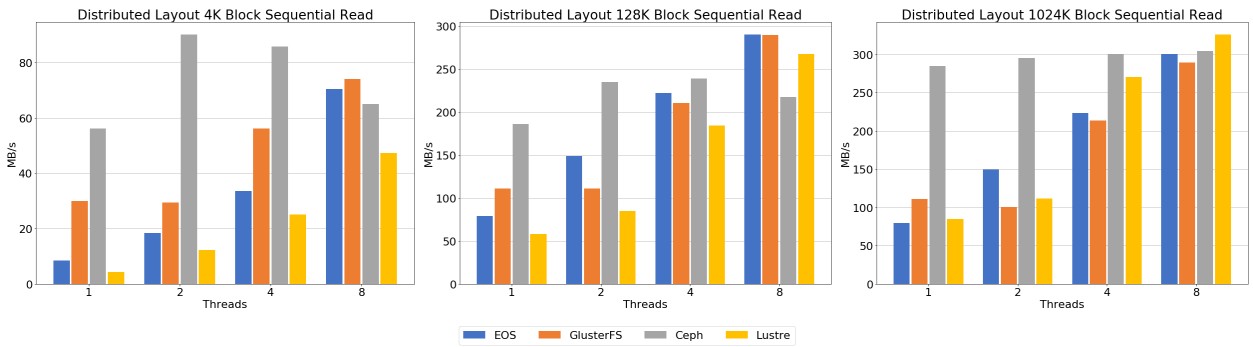

**Figure 10.** Distributed layout—sequential read result.

### 5.1.2. Sequential Write Throughput

Figure 11 shows the result of the sequential write benchmark. Block size has a significant impact on performance in Ceph. The result showed very low throughput compared to the other file systems, except for Lustre at a 4 K block. However, it could be seen that performance was increased compared to the other file systems as the block size increased. EOS showed high throughput with every block size and thread number relative to the other file systems. At a 4 K block, performance increases as the number of threads grows. As shown in the 128 K block results, we can see that if more than one thread is used, the throughput is not increased but saturated. At a 1024 K block, there is stagnation of increased throughput, but overall performance is increased as the number of threads increases. In the case of GlusterFS, the result showed a huge throughput decrease when eight threads were used. As in Ceph, block size has a significant impact on overall performance in Lustre. However, Lustre's performance was relatively poor compared to the other file systems.

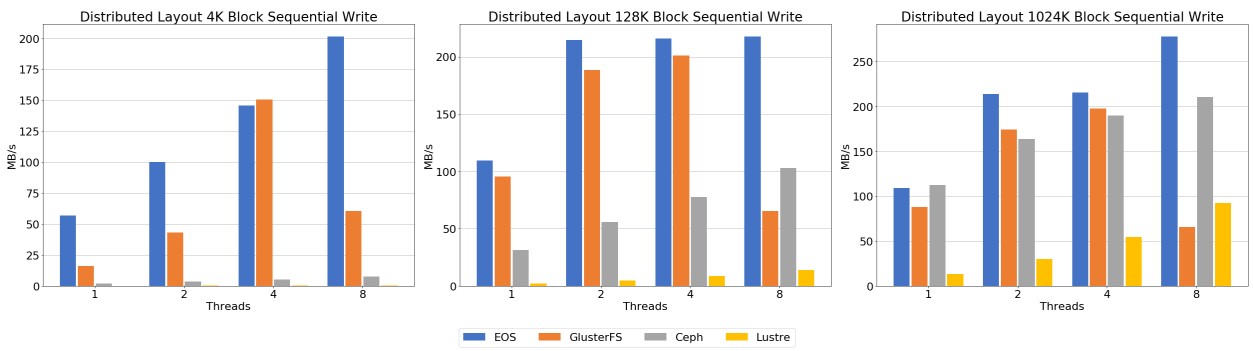

**Figure 11.** Distributed layout—sequential write result.

### 5.1.3. Random Read Throughput

Figure 12 shows the result of the random read benchmark. Ceph showed very high single-thread performance in the 4 K block benchmark, but performance decreased significantly when increasing the number of threads. The results of the 128 K and 1024 K blocks

were very similar and showed higher throughput than the other file systems. However, it should be noted that the increase in threads had a marginal effect on the increase in throughput. In the 4 K block benchmark, the throughput of EOS increases, but decreases when the number of threads is 8, resulting in lower throughput than single-threaded performance. As shown in the 128 K block results, the increase in throughput from two threads or more was insignificant. The result of the 1024 K block shows that throughput increases approximately 70 MB/s as the thread number increases. In the case of GlusterFS, the results showed poor throughput performance compared to the other file systems at a 4 K block. In other block sizes, the throughput increases as the number of threads is increased. At this point, we can see that the performance pattern is very similar to the sequential read graph shown in Figure 10. Luster showed increased throughput when increasing the block size and the number of threads.

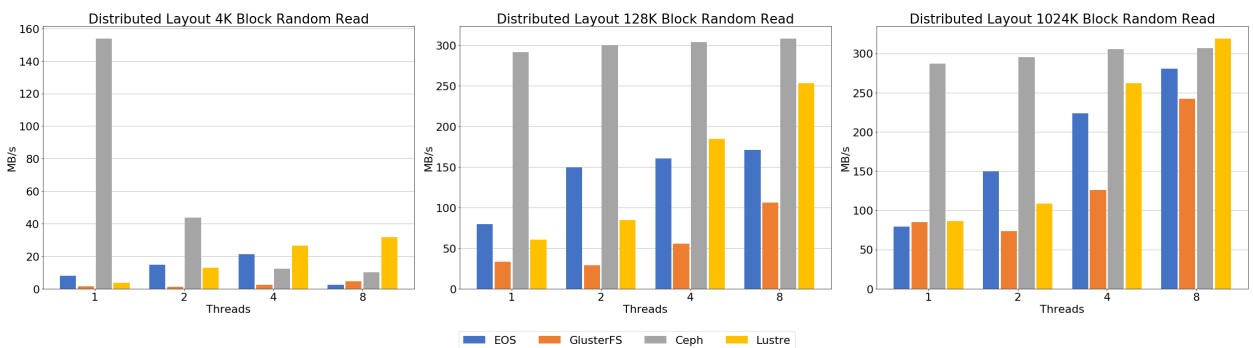

**Figure 12.** Distributed layout—random read result.

### 5.1.4. Random Write Throughput

Figure 13 shows the result of the random write benchmark. Ceph showed a performance increase when the block size and number of threads were increased, similarly to the sequential write result. EOS has high throughput performance in all block sizes compared to the other file systems. At a 4 K block, the result showed a performance decrease at two threads, but high throughput was found with all the other thread options. The 128 K block showed the opposite result to the 4 K block result. At a 1024 K block, we could see that throughput increased, but it was limited at two threads and a fluctuating performance was seen with eight threads. GlusterFS showed similar performance characteristics compared to Ceph, but eight threads caused performance to dramatically decrease, as in sequential write. Luster showed lower throughput performance, as in sequential write, compared to the other file systems, but we saw increases in throughput as block size and number of threads increased.

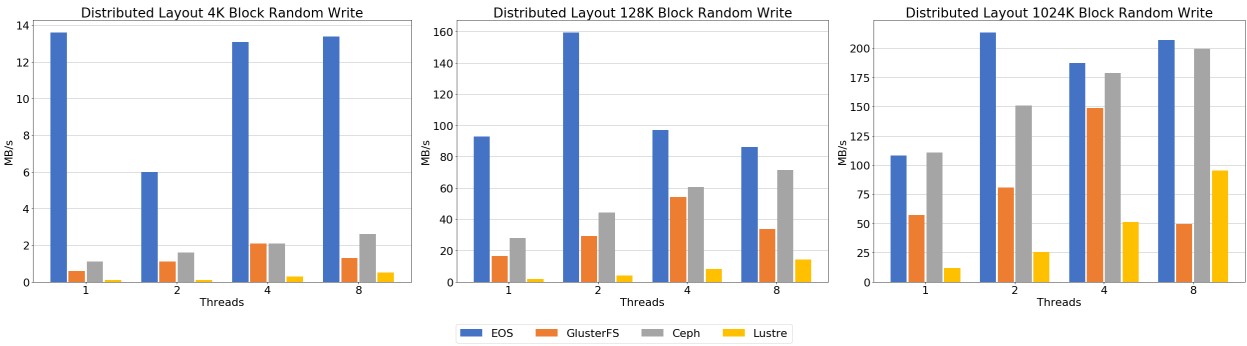

**Figure 13.** Distributed layout—random write result.

### 5.2. RAID 6(4+2)-Like RAIN Layout

5.2.1. Sequential Read Throughput

Figure 14 shows the result of the sequential read benchmark. Ceph's throughput gradually increased from 4 K and 128 K blocks. The throughput increased slightly at the 1024 K block benchmark, but the overall performance stagnated below 200 MB/s on all threads. The EOS 4 K block results showed that throughput increased proportionally to the number of threads and high throughput performance over four threads compared to the other file systems. The 128 K and 1024 K block results showed that throughput performance gradually increased according to the number of threads. In addition, compared to the other file systems, EOS showed higher throughput when EOS used more than four threads. GlusterFS showed similar performance characteristics over all tested blocks. Regardless of the size of blocks, GlusterFS showed the highest performance when using two threads. At a 4 K block, GlusterFS performed more poorly than the other file systems. However, the 128 K and 1024 K block results showed the same throughput performance, outperforming the other file systems.

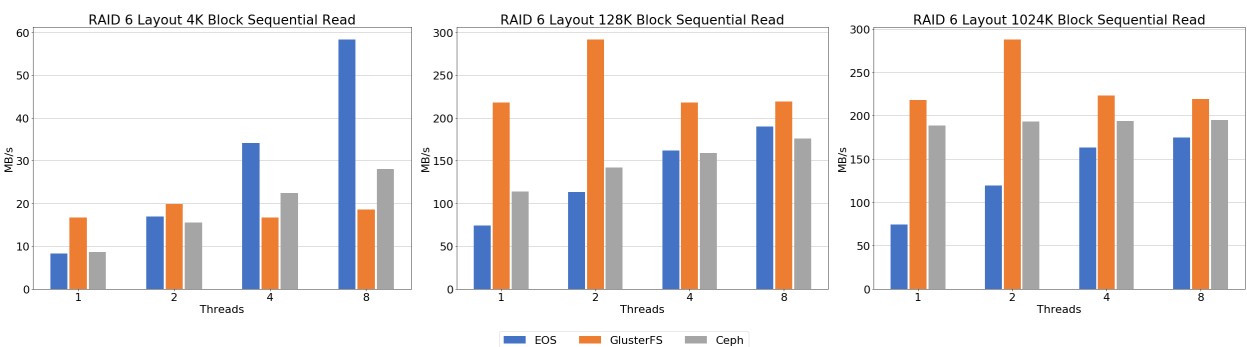

**Figure 14.** RAID 6-like layout—sequential read result.

5.2.2. Sequential Write Throughput

Figure 15 shows the result of the sequential write benchmark. Ceph showed the lowest throughput performance at the 4 K block. We found that increasing the block size had a significant impact on Ceph's throughput. At a 128 K block, a maximum throughput of 24 MB/s and, at a 1024 K block, a throughput of 62 MB/s, were found. In the case of EOS, the result showed a better performance relative to the other file systems. The 4 K block results showed a gradual increase in throughput, but eight threads caused a decrease in throughput compared to four threads. The 128 K and 1024 K blocks resulted in lower single-threaded performance than GlusterFS, but higher performance was found compared to GlusterFS as the number of threads increased. GlusterFS showed low throughput performance in the 4 K block benchmark. As block size increases, GlusterFS shows more improved performance than the previous block size, and 128 K and 1024 K block single-threaded results shows a higher performance than any other file systems. However, GlusterFS showed a dramatic decrease in throughput, unlike the other file systems, with a thread size increase.

5.2.3. Random Read Throughput

Figure 16 shows the result of the random read benchmark. Ceph and GlusterFS showed a decrease in throughput with increased threads. GlusterFS showed slightly higher throughput at 4 K and 128 K blocks, while Ceph showed the highest performance at a 1024 K block. In the case of EOS, the overall processing performance also increased as the number of threads increased, but EOS showed a lower performance than the other file systems.

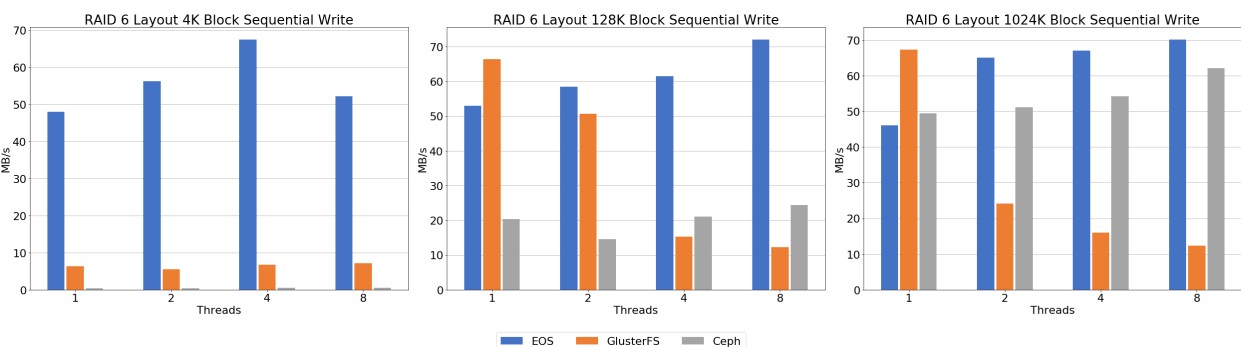

**Figure 15.** RAID 6-like layout—sequential write result.

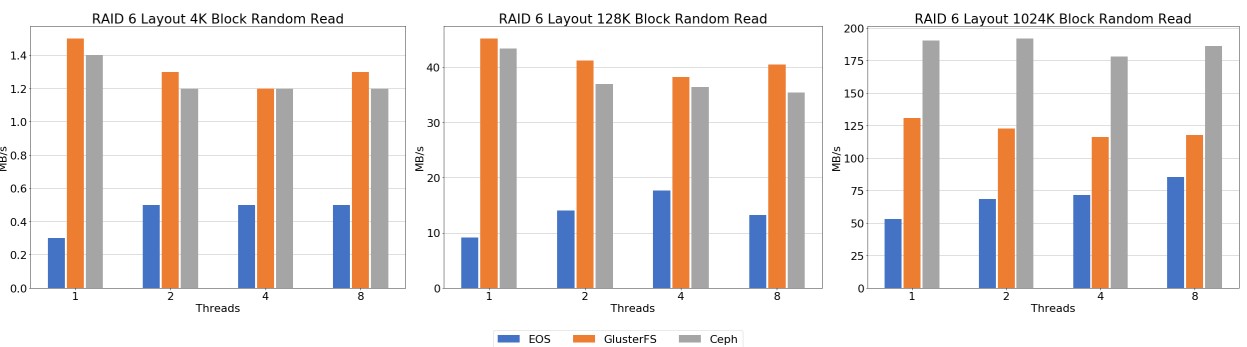

**Figure 16.** RAID 6-like layout—random read result.

### 5.2.4. Random Write Throughput

Figure 17 shows the result of the random write benchmark. Ceph showed enhanced throughput as the block size increased, but the throughput difference was marginal across threads. At 4 K and 128 K blocks, Ceph shows the lowest performance compared to the other file systems, but 1024 K block result shows a much improved performance compared to the smaller block size results. EOS showed the highest throughput at all block sizes compared to the other file systems. The 4 K block result showed a throughput increase with an increasing number of threads. However, the amount of increased throughput declined as the number of threads increased. The 128 K block result showed constant throughput performance throughout all thread numbers. The 1024 K block result showed irregular throughput performance, but the overall performance improved as the number of threads increased. Like Ceph, GlusterFS showed poor performance compared to EOS, and throughput was improved as the block size increased. However, unlike the other file systems, the result of the GlusterFS 1024 K block reveals decreased performance with an increased number of threads.

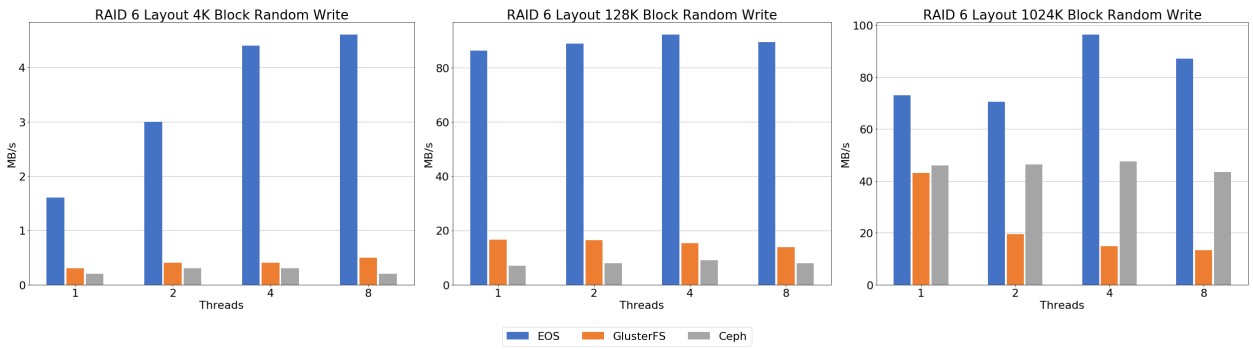

**Figure 17.** RAID 6-like layout—random write result.

## 6. Discussion

In this section, we discuss our evaluation results.

### 6.1. Distributed vs. RAIN

Table 8 describes differences between the distributed and RAID 6 RAIN layout. The distributed layout provides fast and full utilization of configured storage space. It can be used to store large amounts of data. Additionally, the distributed layout is faster than a RAID 6 layout, because it does not perform parity calculations to store data. Therefore, a single disk failure can lead to unrecoverable data loss.

**Table 8.** Comparison between distributed and RAIN layout.

|                              | Distributed | RAIN (RAID 6) |
|------------------------------|:-----------:|:-------------:|
| Additional Parity Calculation | × | ○ |
| Data Redundancy              | × | ○ |
| Storage Throughput Speed     | Fast | Slow |
| Total Disk Capacity Utilization | Full | Partial |

While the RAID 6 layout provides low disk utilization due to parity data compared to the distributed layout, parity data provide data redundancy when part of the storage fails. Additional parity data can recover the failed storage, ensuring data are intact. In addition, the RAID 6 layout has lower throughput than the distributed layout as a result of parity calculations, which can be seen in our results.

### 6.2. Distributed Layout

Sequential and random read benchmarks showed that Ceph's throughput performance was higher than the other file systems in all evaluations. EOS and GlusterFS showed lower performance below four threads, but throughput performance was similar with an increasing number of threads. Lustre showed improved performance with increased block sizes and number of threads, and showed the highest performance when a 1024 K block and eight threads were used.

For writing performance, EOS had higher writing performance than the other file systems. GlusterFS also showed high sequential writing performance, but evaluation with eight threads showed a sharp performance drop. In the case of Ceph, writing throughput performance was significantly improved with increased block size. Lustre had similar performance characteristics to Ceph, but writing throughput was much lower than the other file systems.

### 6.3. RAID 6 RAIN Layout

Unlike the distributed layout, the sequential read performance of GlusterFS was relatively higher than the other file systems, except for a 4 K block. EOS at a 4 K block showed great performance compared to the other file systems when more than four threads were used. Both EOS and Ceph showed similar performance at a 128 K block, but Ceph did not show throughput differences with threads at a 1024 K block.

Random read performance showed that GlusterFS and Ceph had similar performance, but the 1024 K block result showed that Ceph had higher performance than GlusterFS. EOS showed improved throughput performance with increased block size, like Ceph, while throughput was lower than the other file systems.

Both writing benchmark results showed that EOS has great throughput performance. GlusterFS also showed high throughput performance at 128 K and 1024 K blocks and a single thread, but performance sharply dropped with increasing numbers of threads. Ceph showed poor performance with a 4 K block, but throughput performance was significantly improved with increased block size.

## 7. Conclusions and Future Work

As data are ever growing and considered important in scientific data computing, it is important to deploy a proper file systems in such a data-intensive computing environment. When adopting a distributed file system, there are many characteristics which we have to take into account. One of the important characteristics is the I/O patterns shown in data that should be analyzed in such a scientific computing environment. In addition, the reliability results of storage for storing scientific data from the experiment may be more important if the experiments cannot reproduce the results again. Considering such features of data, configuring storage systems has to be carried out.

In this study, we analyzed some well-known distributed file systems in use today and configured a small distributed file system cluster based on commodity hardware and storage disks. Using FUSE clients provided with file systems, we could measure the throughput performance in equivalent conditions with different workloads. Although our experimental cluster environment was very small compared to modern storage solutions, our experimental results identified the characteristics of I/O performance of distributed file systems. In our experiments, it was possible to indicate which file system performed well with different I/O patterns. In addition, we showed that the difference in throughput performance varies with how data is stored in the storage cluster.

Our results and discussion show that a layout has to be chosen depending on how important the data are. In this paper, we discussed two layouts: distributed and RAIN. The distributed layout can use all disk capacities, but does not provide parity calculations, with which disk failure leads to data loss, while RAIN cannot fully utilize the disk capacity, but parity calculations provide an additional layer of protection. With our results, we expect that researchers could select appropriate distributed file systems and layouts according to the importance of data and I/O patterns of research data. We are planning to evaluate the distributed file systems in an improved environment as future work. We will test distributed file systems with detailed parameters, including mixed I/O, metadata operation and latency.

**Author Contributions:** Conceptualization, J.-Y.L. and M.-H.K.; methodology, J.-Y.L., M.-H.K. and S.-Y.N.; software, J.-Y.L.; validation, J.-Y.L. and S.-Y.N.; formal analysis, J.-Y.L. and M.-H.K.; resources, S.-Y.N. and H.Y.; data curation, J.-Y.L.; writing—original draft preparation, J.-Y.L. and M.-H.K.; writing—review and editing, S.A.R.S., S.-U.A., H.Y. and S.-Y.N.; visualization, J.-Y.L.; supervision, S.-Y.N.; funding acquisition, S.-Y.N. All authors have read and agreed to the published version of the manuscript.

**Funding:** This work was supported by the National Research Foundation of Korea (NRF) grant funded by the Korean government (MSIT) (No. NRF-2008-00458).

**Acknowledgments:** The authors would like to extend their sincere thanks to the Global Science Experimental Data Hub Center (GSDC) at the Korea Institute of Science Technology Information (KISTI) for their support of our research.

**Conflicts of Interest:** The authors declare no conflict of interest.

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
