# Peer review of "Performance Evaluations of Distributed File Systems for Scientific Big Data in FUSE Environment"

_electronics, doi:10.3390/electronics10121471_

Round 1

Reviewer 1 Report

The manuscript titled "Performance evaluations of distributed file systems for scientific big data on FUSE environment" investigates and benchmark various distributed file systems such as Ceph, GlusterFS, Lustre, and EOS for the data-intensive environment. The distributed file systems under RAIN structure and FUSE as an accessing environment are configured in this study. The manuscript is well-written however the grammatical revision of the manuscript is necessary. The comments below are suggested by the reviewer for improvement of the manuscript:

  1. The main contribution of the research needs to be represented in the "related work" section. The authors should explain the main reason for using this method by referring to the advantages and disadvantages of previous studies.
  2. The "Background" section should be written before the "related work" section. Or the authors can move this section to the end and call it "Appendix" as the section includes the definition of some technical phrases. 
  3. Please change the title of section 4 to "The experimental setup". 
  4. In figure 17, the EOS results for 128K indicate almost the same values for 1 to 4 threads. How do the authors explain it? Can only 1 thread be used instead of 4 as the achieved results are equal?
  5. In the "conclusion" section, the reviewer's suggestion is to add a sentence about the best "Distributed Layout" based on the achieved results?

Reviewer 2 Report

This paper surveys various distributed file systems such as Ceph, GlusterFS, Lustre and EOS for data-intensive 12 environments. This discussed topic is hot and important. The structure of this paper is well-organized, which is easy to follow. Overall, this paper is a good-quality paper, which will raise wide interests. Congrats! Only one minor issue is that the format of the manuscript and reference should be unified and follow the journal template.
